# Genetic Variation and Composition of Two Commercial Estonian Dairy Cattle Breeds Assessed by SNP Data

**DOI:** 10.3390/ani14071101

**Published:** 2024-04-04

**Authors:** Sirje Värv, Tõnu Põlluäär, Erkki Sild, Haldja Viinalass, Tanel Kaart

**Affiliations:** Institute of Veterinary Medicine and Animal Sciences, Estonian University of Life Sciences, Fr. R. Kreutzwaldi 1, 51006 Tartu, Estonia; tonu.polluaar@emu.ee (T.P.); erkki.sild@emu.ee (E.S.); haldja.viinalass@emu.ee (H.V.)

**Keywords:** Estonian dairy cattle, genomic inbreeding, genetic variation

## Abstract

**Simple Summary:**

The milk production of Estonian dairy cows, with 9529 kg for Estonian Red and 11,394 kg for Estonian Holstein, is the top in Europe. This result has been achieved by more efficient herd management and intensive selection. The latter has a strong effect on genetic variation. Using genome-wide SNP data, it was found that two Estonian dairy breeds have a relatively different genetic variation. Regardless of the smaller population size, the Estonian Red breed had a much lower level of inbreeding than the Estonian Holstein. Compared to the past, the Estonian Red has become more diverse, while the Estonian Holstein has had a decrease in diversity. Similarly, both breeds have negative population trends and gene flow from other populations.

**Abstract:**

The aims of this study were to assess the genomic relatedness of Estonian and selected European dairy cattle breeds and to examine the within-breed diversity of two Estonian dairy breeds using genome-wide SNP data. This study was based on a genotyped heifer population of the Estonian Red (ER) and Estonian Holstein (EH) breeds, including about 10% of all female cattle born in 2017–2020 (sample sizes *n* = 215 and *n* = 2265, respectively). The within-breed variation study focused on the level of inbreeding using the ROH-based inbreeding coefficient. The genomic relatedness analyses were carried out among two Estonian and nine European breeds from the WIDDE database. Admixture analysis revealed the heterogeneity of ER cattle with a mixed pattern showing several ancestral populations containing a relatively low proportion (1.5–37.0%) of each of the reference populations used. There was a higher F_ROH_ in EH (F_ROH_ = 0.115) than in ER (F_ROH_ = 0.044). Compared to ER, the long ROHs of EH indicated more closely related parents. The paternal origin of the genetic material used in breeding had a low effect on the inbreeding level. However, among EH, the highest genomic inbreeding was estimated in daughters of USA-born sires.

## 1. Introduction

The Estonian dairy cattle sector has undergone major changes over the past five decades, including changes in the total number of animals, productivity and the structure of the breeds. According to the Estonian Livestock Performance Recording Ltd. (Tartu, Estonia) [1] data, the population of dairy cows has decreased from 314.1 thousand in 1980 to 80.3 thousand in 2023—that is, by almost 75% within about 10 generations. Within this period, there was a particular decline in the number of animals during the 1990s. Also, the proportions of the breeds have changed significantly. While there were equal proportions of Estonian Red (ER) and Estonian Holstein (EH) breeds in the mid-1980s, at present, the balance is towards the breed with the more productive milk yield (EH), and 89% of cows (70,456 cows) are Holsteins, while only 11% of cows (8924 cows) are Estonian Reds [2,3]. In addition to the decrease in the number of animals, other significant changes have taken place in the Estonian dairy sector during the last two decades. There has been a large decrease in the number of small-scale farms, and at present, the majority of cows are kept in herds of >300 animals, which is in line with more efficient husbandry as herd size has a positive impact on efficiency [4,5]. However, the number of dairy herds in milk recording nationally has decreased from 3211 in 2000 to 351, currently comprising 80,341 dairy cows, of which 97.0% are included in national dairy performance recording. During the same period of time, however, milk production per cow for both breeds increased significantly; currently, it is 9529 kg for ER and 11,394 kg for EH per year [3]. Compared to 1980, there has been a 3-fold increase in milk production in EH and a 2.7-fold increase in ER. As a part of the European red dairy population, the breeders of the ER use genetic material from other European red dairy cattle breeds, including Finnish and Scandinavian breeds. The main foreign components are the Viking Red and the Red Holstein. Less than 20% of the bulls used are born in Estonia. The early ancestry of the ER from the period of the breed’s formation in the late 19th century links the breed to the Danish Red and Angeln breeds. Current EH breeders use Holstein genetic material from around the world. Both red-and-white and black-and-white Holstein genes are accepted in breeding programs. The initial EH population, the Estonian Black and White Cattle, was based on the Dutch Holstein Friesian breed and was renamed the Estonian Holstein in 1997 after intensive “Holsteinization” since the mid-1970s [6,7].

Declines in livestock animal numbers, along with the loss of local breeds, are reducing global biodiversity. In addition to recording pedigree and performance data, the assessment of genetic variability is also important for national genetic resource management [8,9]. The level of inbreeding is an important indicator of the genetic status of a population. Inbreeding has received particular attention in relation to the use of genomic selection and the widespread use of genome-wide SNP data in monitoring breeds [10,11,12,13]. Negative effects of inbreeding on production and fertility have been observed in both Holstein and red breeds [14,15,16,17,18]. Based on genomic studies of cattle breeds, genetic variation and levels of inbreeding vary widely among breeds. Schmidtmann et al. demonstrated a wide range of observed heterozygosity and genomic inbreeding for breeds from Germany, the Netherlands and Denmark [19].

The pedigree-based study of Estonian dairy cows revealed that the average inbreeding coefficient of cows born in 2009–2018 was low (F = 0.023) [20]. However, this value is higher than the F = 0.015 for a large European red dairy cattle population [21] and lower than F = 0.052 found for Dutch–Flemish Holstein–Friesians [11]. Since breeding goals can be achieved by using the best-rated genetic material, breeders prefer to select from a limited number of elite bulls, and inbreeding is expected to increase. Of the Estonian dairy cows born in 2009–2018, the offspring of the five most used bulls make up 10.4% and the offspring of the 20 most used bulls make up 27.6% of the total. The pedigree-based inbreeding coefficient was higher than zero for all the 20 most used bulls (ranging from 0.011 to 0.052). In addition, additive genetic relationship coefficients between the 20 most used bulls were all greater than zero [22].

A previous study of microsatellite variation of 48 *Bos taurus* cattle breeds [23] showed the ER clustering with the Danish Red, Latvian Brown, Latvian Danish Red, Lithuanian Red and Suksun breeds, which was distinct from a Nordic subgroup, including Finnish Ayrshire, Swedish Red and White and the Norwegian Red. It has also been shown that the within-breed diversity of Estonian dairy cattle breeds was at the same level as that of other modern dairy breeds, and the within-breed inbreeding coefficient showed lower inbreeding in the case of the Estonian Holstein (−0.016) than the Estonian Red (0.026) [24].

In relation to long-term sustainability, there is an obvious need for estimates of the genetic diversity of breeds. The start of the genomic era with the introduction of genomic selection in animal breeding requires accurate and up-to-date knowledge of all breeds to minimize undesirable effects and maintain variation. There is a lack of genome-wide data analysis for Estonian cattle breeds. The aim of this study was to assess the genetic diversity of the two Estonian dairy cattle breeds, focusing on genomic inbreeding estimates, and to compare the ER and EH breeds in the context of other European breeds.

## 2. Materials and Methods

### 2.1. Animals and Genotypes

This study was based on single-nucleotide polymorphism (SNP) genotype data of 2265 Estonian Holstein (EH) and 215 Estonian Red (ER) individuals, born in 2017–2020 and sampled for estimation of genomic breeding values in 2020–2021 (R&D Project 616115780002 of the Estonian Dairy Cluster [25]). The sampled heifers represented ca. 10% of yearly primiparous EH and ER dairy cows of the national herd, and the number of heifers from the two breeds corresponds to the breeds’ proportions in Estonia. In order to collect a representative sample to characterize the actual genetic variability of the Estonian dairy cow population, the heifers were randomly selected. EH samples were collected from the 111 largest farms (an average of 20.4 heifers per farm) and ER samples from 36 farms where the proportions of ER were the highest (an average of 6.0 heifers per farm). Pedigree information was not used in animal selection, except for ER heifers; animals with a cumulative percentage of Red Holstein, Brown Swiss and Montbeliarde genes of greater than 50% were not included [26].

In order to study the changes in the genetic structure of Estonian breeds in the last decade and to characterize the Estonian breeds in a wider context, previously genotyped 53 EH and 40 ER unrelated individuals (considering three generations in pedigrees) born in 2006–2013 were included (R&D Project IUT8-2 [27]). In addition, SNP data of nine other European breeds (Brown Swiss, Finnish Ayrshire, French Red Pied Lowland, Dutch Holstein, Jersey, Norwegian Red, Montbeliarde, Red Angus and Western Finncattle, the number of cows *n* = 232) from the publicly available database WIDDE [28] were included. These breeds were selected according to the genetic material introduced into the Estonian dairy population. One more distant breed (Red Angus) was included by a random choice amongst other breeds. All samples were genotyped with largely common SNP panels based on the Illumina BovineSNP50K Beadchip (including EuroG MD microarray versions).

For the breeds’ relationship and structure analyses, current EH data were reduced from 2265 to 45 and current ER data from 225 to 30 by eliminating individuals with high genetic relatedness using PLINK 1.9 [29,30] (--rel-cutoff 0.025). The SNP data from different datasets were merged (--merge). Only SNPs on the 29 autosomes with call rates > 96% were retained. SNPs with unknown chromosomal location and SNPs deviating from HWE (*p* < 0.0001) were excluded. The call rate threshold for individuals was set to the default value of 0.9. Altogether, the initial dataset contained 2805 animals from 11 breeds having 42,747 SNP genotypes. The varied number of samples and SNPs for different analyses are shown (Figure 1).

### 2.2. Breed Relationship and Structure Analyses

Breed relationship and structure analyses were carried out on 400 individuals from 11 breeds. The EH and ER earlier and current samples were treated as separate populations, increasing the total number of populations to 13 (Figure 1). In total, 40,407 SNPs common in earlier and current EH and ER datasets and the European breeds’ dataset were used.

Principal component analysis (PCA) was used to visualize the genetic clustering of animals and the variability between and within populations. The between-breed relationships were measured by F_ST_ values [31] using PLINK 2.0 [29,32], and for the total F_ST_, PLINK 1.9 [29,30] was used.

A model-based ADMIXTURE analysis was performed using STRUCTURE v2.3.4 [33] with the aim of determining the number of ancestral populations of Estonian dairy breeds and identifying the admixture levels of animals at optimal K bases with the selected set of breeds. The determination of the optimal number of genetic populations was based on a combined dataset of 400 individuals with 10,000 burn-in cycles in STRUCTURE followed by 10,000 MCMC repeats for each population number K from 1 to 14. Each value of K was simulated 15 times. Similarities in structure simulations were calculated with CLUMPP 1.1.2 [34]. The optimal K was determined by the ΔK method [35] using Structure HARVESTER web v.0.694 [36] and by maximum log-likelihood value. The STRUCTURE was run with 14,083 SNPs obtained from the PLINK 1.9 LD-based pruner (--indep-pairwise 100,25 and 0.075). The number of SNPs was reduced to produce a set of SNP data that are in approximate linkage equilibrium with each other.

### 2.3. Within-Breed Variation Analysis

The current EH and ER populations were characterized based on all genotyped individuals (*n* = 2265 and *n* = 215, respectively) by calculating the expected and observed heterozygosity and inbreeding coefficients F_HOM_, based on differences between expected and observed homozygosity, as well as F_ROH_, based on runs of homozygosity regions (ROH) of the individual genome. 

The ROHs were determined using PLINK 1.9 [29,30]. We applied the flag criteria as follows: --homozyg-density 150; --homozyg-gap 1000; --homozyg-het 1; --homozyg-kb 1000; --homozyg-snp 30; --homozyg-window-het 1; --homozyg-window-missing 5; --homozyg-window-snp 30. Inbreeding F_HOM_ was obtained from --het. These parameters were used to detect ROHs from 1 Mb length by a minimum number of 30 consecutive homozygous SNPs with at least one SNP per 150 Kb of the ROH. In this study, one heterozygous and five missing genotypes at the scanning window were permitted. The sliding window size, number of homozygous SNPs and SNP density were set to 30 SNPs and 1 SNP per 150 Kb to avoid the underestimation of the ROHs [37], while for the remaining parameters, default values were used. The rather high number of missing genotypes (five) was allowed, but this should have only a limited effect on ROH detection [38].

The genomic inbreeding F_ROH_ calculation was based on the length of the sum of all ROHs per animal and the total length of the autosome covered by SNPs [39]. The length of the autosome genome covered by SNPs was 2,496,104,570 bp in merged data of ER and EH. The inbreeding coefficient F_ROH_ was calculated for ROH > 1Mb.

The ROHs were classified according to the length of the homozygous region as follows: 1–4 Mb, 4–8 Mb, 8–16 Mb, 16–32 Mb and >32 Mb to distinguish homozygosity arising from closer and more distant ancestral generations. 

According to the pedigree information acquired from Estonian Livestock Performance Recording Ltd., the EH samples were divided into nine sire-origin groups, and the ER samples into four sire-origin groups. These groups were formed according to the country of origin of the cows’ sire and were Canada, Switzerland, DSF (Denmark, Sweden, Finland), Estonia, France, Germany, UK, The Netherlands and USA for EH and Estonia, Finland, Denmark and Sweden for ER. Due to unknown paternity, four EH samples were omitted from the following analyses. For both EH and ER breeds, genomic inbreeding in different sire-origin groups was compared with an analysis of variance followed by the Tukey post hoc test. In addition, the effective number of sires and the ratio of the effective number to the actual number of sires were calculated for ER and EH in all sire-origin groups. The effective number of sires in a group was calculated as described by Doublet et al. [40]:$${Ne}_{group}=1/\sum_{i=1}^{n}{\left(\frac{o_{i}}{\sum_{j=1}^{n}o_{j}}\right)}^{2}$$
where *n* is the number of bulls for a given sire-origin group and *o_i_* is the number of offspring of bull *i*.

All mean values in the text are presented with standard deviations.

## 3. Results and Discussion

### 3.1. Genetic Variability and Clustering of Estonian Dairy Breeds in Comparison with Other Selected Breeds

PCA showed that 41.1% of the total genetic variance characterized by the SNP data was explained by the first four principal components (PC): 16.2%, 9.8%, 8.3% and 6.8%, respectively. The location of animals in PC plots revealed strong breed-specific clusters (Figure 2A). PC1 distinguished the Holsteins (formed by two breeds) from red breeds, while Brown Swiss and Jersey were more distant from the other breeds. 

Of the Estonian breeds in this study, both earlier and current ER samples were located in the middle of the other breeds. The current samples had moved slightly towards the Holstein and were more dispersed compared to the other breeds.

The total F_ST_ = 0.089 described a large amount of variation among the breeds, with about 90% of the variation explained by differences between individuals. The pairwise F_ST_ values ranged from 0.008 between current and earlier EH to an extremely high 0.173 between Brown Swiss and Jersey (Figure 2B). A similarly high F_ST_ of 0.168 has been revealed previously between the Traditional Danish Red and the Improved Red [19].

The Estonian Holstein showed a low differentiation from the Dutch Holsteins (F_ST_ values of 0.028 and 0.039 for earlier and current EH, respectively) as well as the ER (F_ST_ in the range of 0.034–0.041 with the lowest F_ST_ between earlier EH and ER and the highest F_ST_ between current EH and current ER). The current ER population showed a less variable F_ST_ range: from 0.010 with the earlier ER to 0.100 with Jersey. Excluding current ER distances from Brown Swiss and Jersey, the current ER population had lower than average pairwise F_ST_ values (mean of 78 pairwise F_ST_ values was 0.085 ± 0.039), and even pairwise F_ST_ values below 0.050 from French Red Pied Lowland, Norwegian Red, Finnish Ayrshire, Western Finncattle and both Estonian Holstein populations. These results are comparable with F_ST_ values ranging from 0.014 to 0.034 among German Red and White Dual-Purpose, Dutch Meuse-Rhine-Yssel and Dutch Deep Red, which share a common history [19].

In the population structure analysis, the optimal number of genetic populations K was eight according to both Evanno’s ΔK method and log-likelihood values. Similarly to PCA, structure analysis revealed breed-specific clusters of Brown Swiss, Jersey, Montbeliarde and Red Angus. However, like the Estonian breeds, at K = 8, all others also displayed admixture, indicating multiple ancestral components with varied contribution rates to their gene pool (Figure 2C). These results are similar to genomic relatedness and admixtures found in several North European [19], Balkan [41], and Eastern and Northern European [42] cattle breeds.

Although model-based clustering showed that both Estonian breeds have a mixed structure, the current EH showed a dominant component (85.3 ± 4.9% of the total), signifying a more homogeneous structure than that found for the current ER (Figure 2D). The highest ancestral component of the current ER specific to Finnish Ayrshire was much lower (37.0 ± 5.8%), and there was a visible change in the proportion of the Holstein component in the current ER compared to the earlier ER sample. The Holstein-related fractions (blue and black in Figure 2C) showed quite high proportions in the current ER (15.4 ± 4.2% and 13.3 ± 5.2%, Figure 2D), reflecting the prevailing trend in dairy breeding. For ER, the Red Holstein was first introduced in 1982 and has been used in numerous herds in recent years to upgrade the population. However, the minor genetic contributions (1.5 to 13.5%) in ER may reflect earlier Angeln and Danish Red introduced in the 1960s and 1970s and together with Brown Swiss in the 1980s [7]. In addition, we included a sample of the Western Finncattle in the set of breed references to provide an example of the genomic pattern characteristic of the local/indigenous population of the region. The common ~10% Western Finncattle fraction found in ER could be construed as the share of ancient ancestry among ER and Western Finncattle. However, we take this result with caution as this share may be explained by current gene flows and needs further investigation. The genetic mixtures found are protrusive in the context of some distinctive breeds. This structure analysis well reflects the ER breeding strategy presented in the breeding program with the aim of producing a healthier, productive dairy breed by using component breeds allowed [7]. As an example of a genetically well-defined breed, the homogeneity of Jersey has been previously demonstrated using microsatellite markers as well as SNP data [43,44].

An increase in homogeneity was seen in EH (Figure 2C). In a short time (2–3 generations), the “Estonian”-specific component increased from 75.5 ± 10.0% to 85.3 ± 4.9%, while the component common to Dutch Holstein decreased from 18.6 ± 3.9% to 10.4 ± 2.8%. This change in EH is consistent with the intensive gene flow from highly selected North American Holsteins. Due to relationship accumulation among Holstein cattle, a decrease in effective population size and an increase in inbreeding have been observed worldwide. The Dutch Holstein Friesian breed has shown an increase in ROH-based inbreeding rate of 2.1% per generation calculated since 2010 [11], and as a result of relationships, the Holstein breed is accumulating inbreeding at the same rate as an idealized population of only 100 individuals [45], confirming the trend observed in EH.

### 3.2. Within-Breed Diversity of Current Estonian Holstein and Estonian Red

The average observed heterozygosity H_obs_ was 0.337 ± 0.010 in the current EH and 0.360 ± 0.011 in the current ER, showing less variation in the predominant breed, the Holsteins. The average F_HOM_ was −0.006 ± 0.030 in the EH and −0.013 ± 0.030 in the ER, reflecting a major heterozygosity excess and more outbreeding in the red population. Also, the average total ROH length of 287.6 ± 67.2 Mb per animal and genomic inbreeding F_ROH_ = 0.115 ± 0.027 for EH clearly exceeded these estimates for ER (109.7 ± 63.0 Mb and 0.044 ± 0.026, respectively) (Figure 3 and Appendix A).

The number of identified ROHs per individual varied from 4 in the case of the ER to 83 in the EH. The total length of determined ROHs was 356.7 Mb per individual in the ER and 578.7 Mb in the EH. A comparison of breeds by ROH lengths and number of ROHs is shown in Figure 3A.

The categorization of ROHs according to the length of homozygous regions revealed a predominance of shorter (<4 Mb) homozygous regions in the genome, accounting for 56.9% in EH and 66.7% in ER ROHs (Figure 3B). In the size class 4-8 Mb, there were 25.3% and 19.8% of ROHs of EH and ER, respectively; in the size class 8-16 Mb, 13.0% and 9.7% of ROHs; and in the >16 Mb ROH class, 4.8% and 3.8% of EH and ER ROHs. The longest ROH (82.6 Mb) was detected in an EH individual. These differences indicate closer parentage in the EH and more distant in the ER. The comparable low proportions of longer ROH classes and lower increase in cumulative F_ROH_ in longer ROH classes indicate the avoidance of close inbreeding in both breeds, but the breed-wise difference in the proportion of shorter ROH (<4 Mb) indicated more inbreeding from earlier generations for ER. Moreover, as known from the history of the breed, the ER has suffered from a decrease in population size for decades, suggesting a genetic bottleneck. Demographic events, population size and genetic management of the breed play a role in the different distribution of ROHs, and genetic bottlenecks should result in a higher number of ROHs per animal, longer ROHs and higher ROH density [46,47,48]. However, in the present study, ER revealed an admixed population rather than a bottleneck, as shown by the low ROH numbers and ROH lengths (Figure 3A), especially compared to the EH individuals.

The data of Schmidtmann et al. showed that F_ROH>4Mb_ across the European red breeds ranged from 0.028 ± 0.018 to 0.155 ± 0.057, while an extremely high percentage (44%) of homozygous regions of the genome were detected at the individual level [19]. This range is similar to our results found for ER and EH. However, it should be taken into account that the results of different studies may not be directly comparable, because the detection of ROHs is based on different parameters (e.g., SNP density, sliding window size, number of missing genotypes, used SNP panels) [38,49,50].

The general finding of differences in genomic inbreeding estimates for ER and EH is consistent with the study results where the effective population size N_e_ calculated from linkage disequilibrium between SNPs was almost twice as high in the Estonian Red than in the Estonian Holstein [51].

### 3.3. Comparison of Intra-Breed Subgroups

Estonian Holstein. Grouping the cows by sire’s country of origin showed that 75% of the eleven sources of genetic material used for EH originated from the USA, Germany or the Netherlands. In addition, joint Danish, Finnish and Swedish bulls had a significant contribution (11% in total) to the current EH population. The number of cows in different sire-origin groups varied from 15 (UK) to 653 (USA; Figure 4). The USA sire group had the highest effective number of sires (Ne_sire_ = 66.9), followed by the German (30.1), joint Danish–Finnish–Swedish (17.1) and Dutch (16.8) sire groups.

The mean number of daughters per sire varied from 4.5 ± 4.9 in the USA to 7.3 ± 9.2 in the German group. The Estonian sires accounted for 7.1% of all 439 sires, with an average of 3.2 ± 2.4 daughters per bull. There were differences in the intensity of the use of genetic material among the sire-origin groups in terms of the ratio of the effective number of sires to the actual number of sires Ne_sire_/N_sire_ (Figure 4). Since the Ne_sire_ is dependent on group size, the ratio Ne_sire_/N_sire_ enables the comparison of groups of different sizes for assessing genetic management. Among the EH sire origin groups, the Ne_sire_/N_sire_ estimate has a wide range (0.21–0.84), with the highest levels (>0.75) occurring in the smaller sire-origin groups. However, in these subgroups (Swiss, French, and British) it is probably due to the use of individual bulls from these countries and does not indicate a more general breeding strategy. In the relatively large group of Dutch origin, the lowest Ne_sire_ and Ne_sire_/N_sire_ values (Figure 4) are mainly caused by the intensive use of one bull (20% of daughters). The overall Ne_sire_ of EH was 139 with Ne_sire_/N_sire_ = 0.32.

The genomic inbreeding F_ROH_ estimates ranged from 0.101 ± 0.011 in the daughter group of Great Britain bulls to 0.124 ± 0.026 in the daughter group of the USA bulls. The USA group was significantly different (*p* < 0.05) from all others except the Canadian and the low-number French groups. The daughters of sires born in Estonia had a significantly (*p* < 0.05) lower F_ROH_ compared to the USA and Canadian sire groups (Figure 4). Only the USA group had a positive mean F_HOM_ value, while the other groups had negative mean F_HOM_ values, with the Netherlands sire-origin group having the lowest mean F_HOM_ value (Appendix A).

In addition, 4.6% of the total number of Estonian Holsteins were Red and White Holstein individuals in six sire-origin groups (Canadian, Swiss, German, Danish, Dutch and American). Compared to the Black and White Holsteins, individuals of the red variety showed lower average inbreeding (0.088 ± 0.026 vs. 0.117 ± 0.026), reflecting the dissimilar genetic backgrounds of the Black and White Holsteins and Red Holsteins used.

Estonian Red. Four sire-origin groups were established: Danish and Finnish paternity dominated (40.9% and 35.8%, respectively); 14.8% of individuals were of Estonian origin, and 8.5% had Swedish origin. In addition, among the maternal grandsires of the studied ER cows, Danish parenthood predominated over Estonian.

The Ne_sire_ values of the Danish, Estonian and Finnish subgroups were similar, varying from 12.3 to 14.7, and the Ne_sire_ was the lowest (3.3) in the smallest group of Swedish sire origin. The Estonian-origin sires had a lower number of daughters resulting in a high ratio of Ne_sire_/N_sire_ = 0.77. The overall Ne_sire_ for ER was 39, and Ne_sire_/N_sire_ = 0.63. This estimate appears much higher than found for Estonian Holstein (0.32), indicating more intensive breeding in EH, which is in accordance with differences found in inbreeding levels in both ER and EH.

Among the ER sire-origin groups, only marginal differences in genomic inbreeding F_ROH_ were found (Figure 4). However, despite the rather narrow (Northern European) paternal geographical origin, the modern Estonian Red breed shows genetic heterogeneity, as was evident from the admixture analysis (Figure 2). This could be explained by the fact that the genetic differentiation among the Red is higher than that among Holstein national populations. Compared to EH, the sire’s country of origin used in ER reflects gene flow from different populations by varied rates of genes from Brown Swiss, Angler, Montbeliarde, Norwegian Red, Red Holstein and Viking Red. Variations in F_ST_ values are presented in the current study above (Figure 2B) and in [19].

## 4. Conclusions

Estonian Red and Estonian Holstein displayed different patterns of within-breed variation. The genomic data showed that regardless of the smaller population size, the Estonian Red had a much lower level of inbreeding than the Estonian Holstein. Compared to the past, the ER has become more diverse while the EH has had a decrease in genetic diversity. Similarly, both breeds experience ongoing negative population trends and gene flow from other populations. The distinctions in population genetic characteristics are in accordance with the varied range of breeding components used for interbreeding in Estonian Red and herd management accumulation into large-scale farms by intensive use of genetic material in the Estonian Holstein. 

## Figures and Tables

**Figure 1 animals-14-01101-f001:**
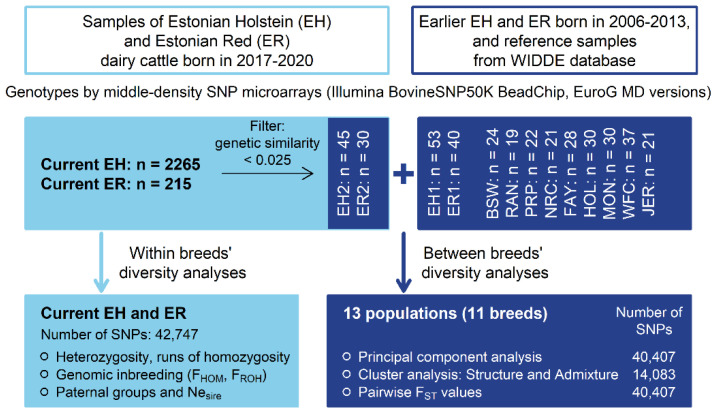
Schematic presentation of study design and analyses (EH1—earlier Estonian Holstein; EH2—current Estonian Holstein; ER1—earlier Estonian Red; ER2—current Estonian Red; BSW—Brown Swiss; RAN—Red Angus; PRP—French Red Pied Lowland; NRC—Norwegian Red; FAY—Finnish Ayrshire; HOL—Dutch Holstein; MON—Montbeliarde; WFC—Western Finncattle; JER—Jersey).

**Figure 2 animals-14-01101-f002:**
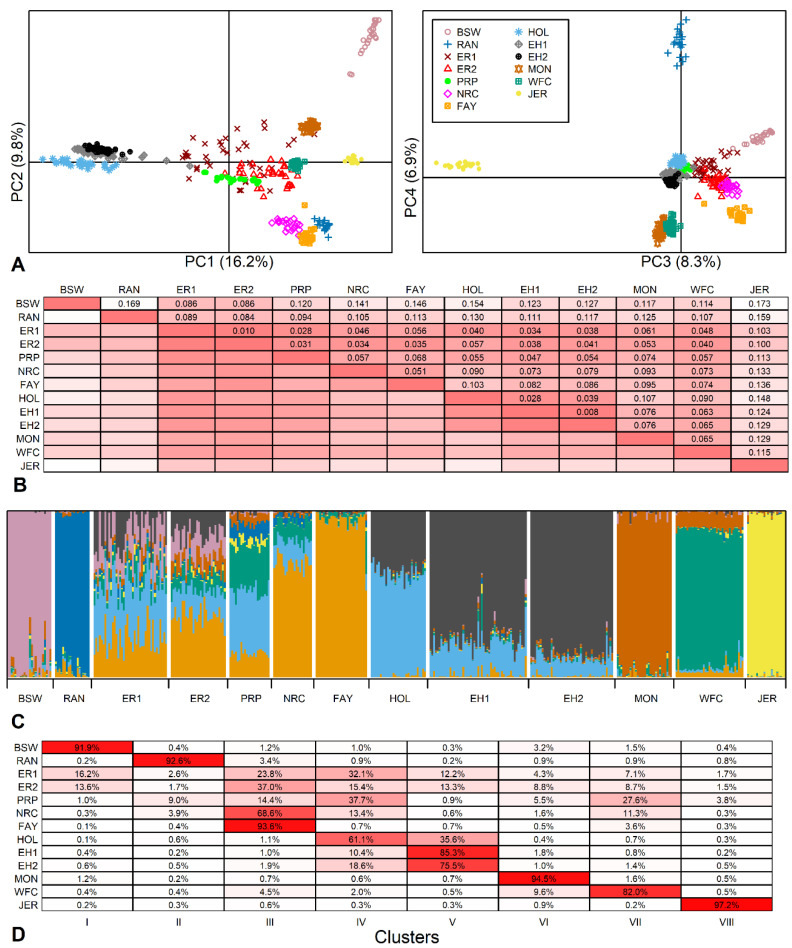
Genetic variability between breeds. (**A**) Location of 400 animals from 13 breed samples (11 breeds) according to the principal component (PC) analysis (BSW—Brown Swiss; RAN—Red Angus; ER1—earlier Estonian Red; ER2—current Estonian Red; PRP—French Red Pied Lowland; NRC—Norwegian Red; FAY—Finnish Ayrshire; HOL—Dutch Holstein; EH1—earlier Estonian Holstein; EH2—current Estonian Holstein; MON—Montbeliarde; WFC—Western Finncattle; JER—Jersey) based on 40,407 SNPs. (**B**) Pairwise F_ST_ values, more intense color indicates more closely related populations (smaller F_ST_ values). (**C**) Individual clustering of animals using admixture analysis of 14,083 SNPs. K = 8 was selected according to Evanno’s ΔK and likelihood function values as the number of clusters best fitting the data. (**D**) The rates of admixture clusters (inferred ancestral populations) in studied populations, more intense color indicates higher admixture ratio of a particular breed (in rows) to the selected cluster (in columns).

**Figure 3 animals-14-01101-f003:**
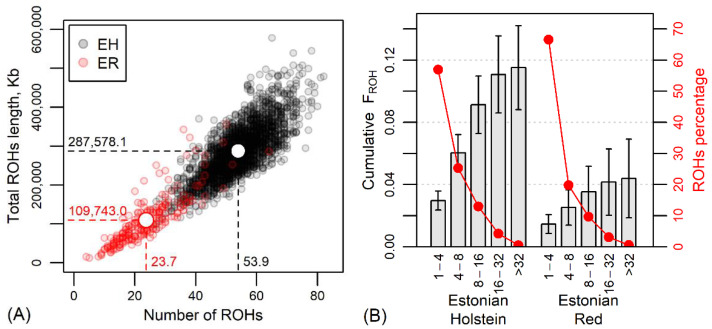
(**A**) The relationship between the number of runs of homozygosity regions (ROHs) and the total ROH length in current Estonian Holstein (EH) and Estonian Red (ER) dairy cows. White circles, dashed lines and numerical values indicate the mean values by breeds. (**B**) Cumulative genomic inbreeding F_ROH_ (±standard deviation) and percentage of ROHs according to the length of homozygous regions (Mb) and breed.

**Figure 4 animals-14-01101-f004:**
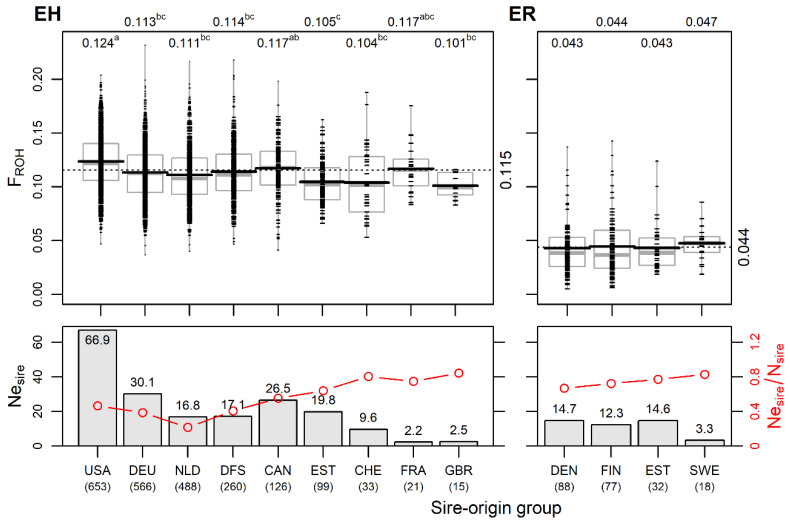
Differences in genomic inbreeding F_ROH_ and in effective number of sires by sire-origin groups in Estonian Holstein (EH) and Estonian Red (ER). Upper figures: F_ROH_ by sire-origin groups of current EH (left panel) and ER (right panel) cows. Small dashes mark individual cows; horizontal black and grey lines mark the mean and median, respectively; dotted line and numerical values on the right side indicate the overall mean of breed; numerical values above the figure indicate mean values by sire-origin groups; means without a common superscript letter are statistically significantly different (*p* < 0.05, Tukey post hoc test). Lower figures: effective number of sires Ne_sire_ (bars) and the ratio of the effective number to the actual number of sires Ne_sire_/N_sire_ (red lines) by sire-origin groups in breeds. Below country names, the total numbers of cows in sire-origin groups are shown (USA—United States of America; DEU—Germany; NLD—Netherlands; DFS—Denmark; Sweden; Finland; CAN—Canada; EST—Estonia; CHE—Switzerland; FRA—France; GBR—Great Britain; DEN—Denmark; FIN—Finland; SWE—Sweden).

## Data Availability

The data presented in this study are available on request from the corresponding author. The data from the WIDDE database are publicly available at http://widde.toulouse.inra.fr/widde/ (accessed on 15 January 2024).

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
