# Peer review of "Genetic Variation and Composition of Two Commercial Estonian Dairy Cattle Breeds Assessed by SNP Data"

_animals, 2024, doi:10.3390/ani14071101_

Round 1

Reviewer 1 Report

Comments and Suggestions for Authors

The paper offers a genomic characterization of the two main Estonian dairy cattle breeds, delving into genomic structure, relationships with other European breeds, inbreeding, and within-breed differentiation. The topic holds potential interest for readers, and I have only minor concerns regarding content, as noted below and in the attached PDF file. However, I recommend improving the writing quality; although the manuscript is overall grammatically correct, the writing style occasionally reduces clarity, particularly in the introduction. 

Introduction:

·        While the content is adequate, enhancing the English and refining the style could enhance readability and reduce redundancies, as the current presentation can be challenging to follow.

·        There is a lack of reference for data on changes in the Estonian cattle population. If all data are derived from The Estonian Animal Recording Yearbooks, it may be beneficial to specify this at the beginning of the related paragraph, directly in the main text. Additionally, references for lines 35-75 are still absent.

·        The introduction appears somewhat lengthy, and maybe some information might be better integrated into the discussion section.

Results/Discussion:

·        When presenting means, it is advisable to include a measure of dispersion, such as standard deviation. Also, generally, mean and standard deviation or other measures of central tendency and dispersion provide a more comprehensive representation of the data compared to the range.

·        It would be beneficial to include a bar plot depicting FROH by ROH length class for both breeds to facilitate comparison.

Conclusions/General:

·        To better support the concluding sentence, a brief paragraph explaining known differences in management between the two breeds could be included.

Comments on the Quality of English Language

Please, see "Comments and Suggestions for Authors
" section.

Author Response

Thank you for the comments and recommendations. We revised the manuscript and made corrections concerning the language. We added the information of the Estonian Animal Recording Yearbooks as suggested and added references concerning the data on the breeds and their changes in introduction. We also removed some specific information from the Introduction.

In Results and Discussion section the variability measures were missing in several places and have now been included. Now all mean values are presented with standard deviations (this information is now mentioned also in Materials and Methods). We also constructed a plot showing cumulative FROH and ROH proportions by the ROH size classes (Figure 3B).

We rewrote the last concluding sentence and included sentence discussing the connection between results of present study and actual breed management into Results and Discussion.

Reviewer 2 Report

Comments and Suggestions for Authors

Journal: Animals (ISSN 2076-2615)

Manuscript ID: animals-2906757

Type “Genetic Variation and Composition of Two Commercial Estonian Dairy Cattle Breeds Assessed by SNP Data”

Mayor Comments

In general, the manuscript is written properly, even though several areas could be improved.

Section 2. Material and Methods.

2.1.- the authors must clarify if the samples were taken to this study or if they comes from a breeder or national databank of information. If samples were taken to this study, please provide the animal permission from the animal care committee in the university. Please indicate more relevant details of the sampled animals, if this information could be useful to the aim of the study.

I would suggest that the authors be comprehensive in explaining the methods implemented by the mentioned software. The authors are just mentioning the commands with further context.

The authors did three analyses. Two of them with 40,407 SNP and one with 14,083. The authors make the question. What happened if all three had been carried out with the same number of SNPs? Could the authors include this information?

Please update the discussion of the results, taking into account the comments on the material and methods.

The conclusion looks like an axiom.

The authors mention “There are some differences in breeding practices (genetic management, differences in contribution of bulls used, effective number of sires), and farming (breed fragmentation/keeping in different herd sizes) between the breeds that affect current population on global and local levels”. In the study, the breeding practices and farming details were not included. How the authors can include it in the conclusion?

Author Response

Thank You for indicating to some lack of information considering sampling data. We utilized genotype data from previous projects; no samples were collected for the current study. We added information concerning the data sources and sampling.

We also added some more explanations about applied algorithms into Materials and Methods. The reason for using different numbers of SNPs in different analyzes is due to the different mathematical assumptions of these analyses. In the admixture analysis, the adopted likelihood model assumes unlinked loci. Since the theoretical assumptions are almost never 100% valid in real life, most methods are applied approximately. In present analyses we took into account the recommendations given in software manual and reduced the number of markers based on the linkage between them (one of the SNPs in case r2>0.075 between loci was removed). As in other applied analyses there were no such mathematical assumptions, the number of SNPs was not reduced to avoid underestimation of the genetic variability and relatedness.

We improved slightly the Results and Discussion as well the Conclusion section. We removed the discussion/conclusion about farming details as we are lacking the publications about them (the analyses are going on and the publication is still in preparation face), we also added some sentences about breeds’ management (as proposed in breeding programs).

Round 2

Reviewer 2 Report

Comments and Suggestions for Authors

I have no more comments on the manuscript.

Author Response

Thank you for your time and for your comments previously. We are happy that you find the revised manuscript worthy of publication in Animals.